# Effect of Hypoxia in the Transcriptomic Profile of Lung Fibroblasts from Idiopathic Pulmonary Fibrosis

**DOI:** 10.3390/cells11193014

**Published:** 2022-09-27

**Authors:** Yair Romero, Yalbi Itzel Balderas-Martínez, Miguel Angel Vargas-Morales, Manuel Castillejos-López, Joel Armando Vázquez-Pérez, Jazmín Calyeca, Luz María Torres-Espíndola, Nelly Patiño, Angel Camarena, Ángeles Carlos-Reyes, Edgar Flores-Soto, Guadalupe León-Reyes, Martha Patricia Sierra-Vargas, Iliana Herrera, Erika Rubí Luis-García, Víctor Ruiz, Rafael Velázquez-Cruz, Arnoldo Aquino-Gálvez

**Affiliations:** 1Facultad de Ciencias, Universidad Nacional Autónoma México (UNAM), Mexico City 04510, Mexico; 2Laboratorio de Biología Computacional, Instituto Nacional de Enfermedades Respiratorias Ismael Cosío Villegas (INER), Mexico City 14080, Mexico; 3Laboratorio de Biología Molecular, Departamento de Fibrosis Pulmonar, Instituto Nacional de Enfermedades Respiratorias Ismael Cosío Villegas (INER), Mexico City 14080, Mexico; 4Departamento de Epidemiología y Estadística, Instituto Nacional de Enfermedades Respiratorias Ismael Cosío Villegas (INER), Mexico City 14080, Mexico; 5Laboratorio de Biología Molecular de Enfermedades Emergentes y EPOC, Instituto Nacional de Enfermedades Respiratorias Ismael Cosío Villegas (INER), Mexico City 14080, Mexico; 6Division of Pulmonary, Critical Care and Sleep Medicine, Department of Internal Medicine, Davis Heart and Lung Research Institute, The Ohio State University, Columbus, OH 43210, USA; 7Laboratorio de Farmacología, Instituto Nacional de Pediatría (INP), Mexico City 04530, Mexico; 8Unidad de Citometría de Flujo (UCiF), Instituto Nacional de Medicina Genómica (INMEGEN), Mexico City 14610, Mexico; 9Laboratorio de HLA, Instituto Nacional de Enfermedades Respiratorias Ismael Cosío Villegas (INER), Mexico City 14080, Mexico; 10Laboratorio de Onco-Inmunobiología, Departamento de Enfermedades Crónico-Degenerativas, Instituto Nacional de Enfermedades Respiratorias Ismael Cosío Villegas (INER), Mexico City 14080, Mexico; 11Departamento de Farmacología, Facultad de Medicina, Universidad Nacional Autónoma de México (UNAM), Mexico City 04510, Mexico; 12Laboratorio de Genómica del Metabolismo Óseo, Instituto Nacional de Medicina Genómica (INMEGEN), Mexico City 14610, Mexico; 13Departamento de Investigación en Toxicología y Medicina Ambiental, Instituto Nacional de Enfermedades Respiratorias Ismael Cosío Villegas (INER), Mexico City 14080, México; 14Laboratorio de Biología Celular, Departamento de Fibrosis Pulmonar, Instituto Nacional de Enfermedades Respiratorias Ismael Cosío Villegas (INER), Mexico City 14080, Mexico

**Keywords:** hypoxia inducible factors, microarray, IPF fibroblasts

## Abstract

Idiopathic pulmonary fibrosis (IPF) is an aging-associated disease characterized by exacerbated extracellular matrix deposition that disrupts oxygen exchange. Hypoxia and its transcription factors (HIF-1α and 2α) influence numerous circuits that could perpetuate fibrosis by increasing myofibroblasts differentiation and by promoting extracellular matrix accumulation. Therefore, this work aimed to elucidate the signature of hypoxia in the transcriptomic circuitry of IPF-derived fibroblasts. To determine this transcriptomic signature, a gene expression analysis with six lines of lung fibroblasts under normoxia or hypoxia was performed: three cell lines were derived from patients with IPF, and three were from healthy donors, a total of 36 replicates. We used the Clariom D platform, which allows us to evaluate a huge number of transcripts, to analyze the response to hypoxia in both controls and IPF. The control′s response is greater by the number of genes and complexity. In the search for specific genes responsible for the IPF fibroblast phenotype, nineteen dysregulated genes were found in lung fibroblasts from IPF patients in hypoxia (nine upregulated and ten downregulated). In this sense, the signaling pathways revealed to be affected in the pulmonary fibroblasts of patients with IPF may represent an adaptation to chronic hypoxia.

## 1. Introduction

Idiopathic pulmonary fibrosis (IPF) is a chronic, progressive, and lethal disease associated with aging [1]. Its pathophysiology consists of damage to the epithelial cells with an aberrant response characterized by the production of several mediators for the fibroblast’s activation. As a result, lung parenchyma is replaced by exacerbated extracellular matrix deposition, which interrupts the oxygen supply [2,3]. Although hypoxia signaling has been reported to be active in the lungs of IPF patients, the role of hypoxia in the pathogenesis of IPF remains unclear [4,5,6].

Hypoxia is a stress condition that influences cell fate by modifying numerous circuits. In this context, especially in the “fibroblast foci”, the main histopathological characteristic of these patients are the mechanisms of hypoxia adaptation that result in profibrotic feedback signaling, which could perpetuate a fibrotic state [7]. For example, hypoxia and the hypoxia transcription factors (HIF-1α and 2α) are involved in the differentiation of myofibroblasts, extracellular matrix deposition, and alteration in the cell cycle [8,9]. In our previous work, IPF fibroblasts show a particular adaptation to hypoxia because they overexpress the alpha 1 and 2 subunits but not subunit 3 (a negative regulator) of HIF, suggesting a hyperactivation of this pathway even in the presence of oxygen [9]. Although these transcription factors are altered, the impact of hypoxia on the transcriptomic profile has not been determined.

Transcriptomic analysis allows us to define the stimulus and landscape of altered circuits [10]. Therefore, we consider that an analysis of the complete transcriptome under hypoxic conditions could lead us to elucidate the possible mechanisms involved in the development of IPF.

## 2. Materials and Methods

### 2.1. Human Lung Fibroblasts

The bioethics committees of Instituto Nacional de Enfermedades Respiratorias Ismael Cosío Villegas (INER) approved this protocol (B29-20; Date: 11 November 2020). The donation of samples (biopsy) from patients and donors was performed following the relevant guidelines and regulations with the informed consent of all participants, as previously described [9]. IPF was diagnosed following the Interstitial Lung Disease Program of the INER according to the ATS/ERS/ALAT guidelines [11].

Cell culture was performed using Ham’s F-12 medium (Gibco, Waltham, MA, USA) supplemented with 100 U/mL of penicillin, 100 µg/mL of streptomycin, 2.5 mg/mL of amphotericin B, and 10% FBS (Gibco Cat No. 11550356), at 37 °C, in an atmosphere of 95% air and 5% CO_2_.

Primary or commercial cell lines from patients with IPF (F1, F2, and F3) and controls (C1, C2, and C3) were used; cell demographics are shown in Table 1. All experiments were performed in triplicate.

### 2.2. Hypoxia

For the experiments under hypoxic conditions, the fibroblasts were cultured in a 60 mm culture dish and subsequently transferred and maintained in a modular incubation chamber (Model MIC-101 of the brand Rothenberg Inc., San Diego, CA, USA), at 37 °C, for 48 h, in a humidified atmosphere with the following mixture of hypoxic gases: 1% O_2_, 5% CO_2_, and balanced with N_2_. An oxygen analyzer (Teledyne Electronic Technologies 60T) with an oxygen sensor (OOM105 from EnviteC-Wismar GmbH, Wismar, Germany) was used to monitor oxygen concentration.

### 2.3. Total RNA Extraction

Total RNA was isolated with RNeasy Mini Kit (Qiagen, Redwood City, CA, USA) according to the manufacturer’s protocol. Approximately 8 × 10^5^ cells were used; once washed twice with Phosphate Buffered Saline (PBS) in Petri plates of 60 mm, 600 µL of RLT buffer was added per plate. They were detached with the help of a police rubber, then were transferred to a tube, and were homogenized for at least 45 s at maximum speed; afterwards, 70% ethanol was added in a 1:1 ratio and homogenized well; 700 µL of this mixture was transferred to an RNeasy Mini spin column with a 2 mL collection tube and centrifuged at 9000 rpm/30 s, discarding what was obtained in the tube. Subsequently, 700 µL of RW1 buffer was added to the RNeasy Mini spin column, and it was centrifuged at 9000 rpm/30 s. Again, what was collected was discarded. Later, 500 µL of Buffer RPE was added to the RNeasy spin column, and it was centrifuged at 9000 rpm/30 s. This collection tube was discarded together with what was obtained; a new tube was placed for RNA storage, and 40 µL of RNase-free water was added directly to the spin column membrane and centrifuged at 900 rpm/1 min to elute RNA.

Extracted RNA was quantified spectrophotometrically with NanoDrop ND-1000 (Thermo Fisher Scientific, Wilmington, DE, USA). RNA quality was assessed with an Agilent 2100 Bioanalyzer (Agilent Technologies, Santa Clara, CA, USA).

### 2.4. Microarray Hybridization

Isolated total RNA was amplified, labeled, and hybridized using the Clariom D platform, which detects >540,000 transcripts, following the manufacturer’s instructions (Thermofisher, Cat. 902923, Santa Clara, CA, USA). Briefly, the Affymetrix GeneChip^®^ WT PLUS Reagent Kit (Santa Clara, CA, USA) was used for cDNA preparation and biotin labeling. cRNA was purified using an Affymetrix magnetic bead protocol. The Affymetrix GeneChip Hybridization, Wash, and Stain kits were used for array processing. Arrays were incubated for 16 h in an Affymetrix GeneChip 645 hybridization oven at 45 °C with rotation at 60 rpm. The chips were subsequently scanned with an Affymetrix GeneChip Scanner 3000. Raw data were analyzed using Affymetrix Expression Console and Transcriptome Analysis Console software prior to downstream analysis.

### 2.5. Analysis of Differential Gene Expression

Microarray data can be downloaded from the https://figshare.com/projects/Hipoxia/141623. The microarray data were analyzed using R software version 4.1.0 [12], and Bioconductor version 3.13 [13]. Quality analysis was performed using affycoretools package version 1.64 [14]. We normalized the data using RMA (Robust Multiarray Average) [15] to minimize the non-biological variation in signal intensities.

To identify significant differences between gene expression in each condition, we selected the contrasts specified in Appendix A. In general, we performed two different analyses. First, we performed a contrast comparing all the IPF fibroblast primary cell lines vs. all the control fibroblast cell lines, and second, we compared IPF vs. controls in each cell line individually. All data were analyzed using the limma package version [16] using a linear model based on Bayes empirical method [17]. Gene annotation was performed using package pd.clariom.d.human version 8.8 [18]; here, we present only those genes that had a NM accession. Genes were considered statistically significant with higher *p*-values (adjusted *p*-value < 0.05), and logFC > 1 or logFC < −1.

### 2.6. Data Visualization

Ggplot2 version 3.3.5 [19] was used, EnhancedVolcano package version 1.10 [20], ComplexHeatmap package version 2.8.0 [21], and VennDetail Shiny App (http://hurlab.med.und.edu:3838/VennDetail/) [22], for PCAs, volcanos, heatmaps, and Venn diagrams, respectively. The final versions of the figures were edited using Adobe Illustrator program.

### 2.7. Enrichment Pathways Analysis and Networks Generation

The pathways and networks were generated using the differentially expressed genes using QIAGEN Ingenuity Pathway Analysis, version 2000–2022 (https://digitalinsights.qiagen.com/IPA) [23]. An enrichment analysis pathway was also performed on the genes obtained through the Venn diagrams using EnrichR with the Gene Ontology Biological Process [24,25,26].

## 3. Results

### 3.1. Transcriptional Response to Hypoxia in Control and IPF-Derived Lung Fibroblasts

To evaluate the effect of hypoxia on control lung fibroblasts and lung fibroblasts from IPF patients, the cells were cultured for 48 h in a hypoxic chamber at 1% O_2_. For this, three lines of fibroblasts from healthy lungs were used as controls (C1, C2, and C3), and three primary lines of lung fibroblasts from patients with IPF (F1, F2, and F3) were used to address the heterogeneity of cell lines and the variability in the hypoxic stimuli. Subsequently, the samples were analyzed with the Clariom D platform (Figure 1).

Analysis was carried out in two ways: first, by the bulk of cell lines by condition (normal or fibrotic) and, second, by determining the specific response for each cell line. It is essential to mention that when the dispersion was analyzed in the PCAs, control or fibrotic (C or F) cell lines were clustered together in their respective condition group (Appendix A). Cell lines (C or F) also cluster together, suggesting that gene expression changes are similar (Appendix A).

Analysis by condition revealed a strong transcriptional response to hypoxia in control fibroblasts, which showed 1006 differentially expressed genes (DEGs) (Figure 2A,C). In sharp contrast, IPF-derived fibroblasts displayed a modest response with only 241 DEGs (Figure 2B,D). This phenomenon could be explained because IPF fibroblasts were previously exposed to this stimulus, suggesting preadaptation to this condition [9]. Despite the difference in number of genes, both IPF and controls depict the same biological pathways and processes according to Gene Ontology analysis (GO): cellular response to hypoxia (GO:0071456) including the cellular response to decreased oxygen levels (GO:0036294) (Figure 2E,F and Appendix A).

Network analysis using Ingenuity Pathway Software (IPA) shows a transcriptomic landscape in normal fibroblasts during hypoxia revealing metabolic changes regulated by HIF-1α (Figure 3A,B). These changes showed a central role of HIF-1α in YTH N6-Methyladenosine RNA Binding Protein 2 (YTHDF2)-mediated cell cycle regulation which promotes mRNA decay during the cell cycle in somatic reprogramming (Figure 3A). Consistent with that, TP53, a master cell cycle regulator implicated in aging, senescence, and fibrosis, was upregulated (Figure 3C). The radial diagram of TP53 and its targets reveals its central role in the regulation of autophagy-related processes (ATG4a and ATG4b), topological stress (TOP2B), glycolysis (TIGAR), and insulin-like growth factor regulation (IGF) and transport and uptake by insulin-like growth factor binding proteins (PAPPA). In summary, controls have a complex network involving more genes in the hypoxia adaptation and antagonistic mechanisms such as autophagy regulation. IPA also identified alterations in the AKT signaling in IPF; AKT can negatively regulate autophagy in the opposite direction to controls (Figure 3D).

To further evaluate possible directional consequences and inferred upstream regulator activity in HIF-1α signaling in hypoxia in both control and fibrotic fibroblasts, we used the IPA tool Molecular Predicted Map on our data set, where it is also shown that the controls have greater complexity in the signaling pathways (Figure 4 and Figure 5).

### 3.2. Shared Hypoxic Genes Float in a Sea of Doubts

The Venn diagram of Figure 6A shows us that there are 346 DEGs shared between control cell lines, and Figure 6B shows that there are 54 DEGs shared between lung fibroblasts from patients with IPF in hypoxia.

The number of unique DEGs that are not shared with any other cell lines is heterogeneous in number; the control lines C1, C2, and C3 have 1412, 209 and 540 genes, respectively (Figure 6A), while in the three lines of patients with IPF, we can observe that F1, F2, and F3 possess 845, 49, and 274 genes, respectively (Figure 6B).

The pathways in the group of fibroblasts from controls are related to the epithelium development, differentiation, and pathways involved with several metabolites (Figure 6C). Due to the experimental time (48 h) varying from that published in previous studies (6 h and 12 h), some genes related to metabolism differed. In IPF fibroblasts, enrichment analysis suggested that angiogenesis-related pathways are involved (Figure 6D).

### 3.3. The Transcription Signature of Hypoxia in IPF Fibroblasts

A third Venn diagram was constructed to identify the transcriptional signature with dysregulated genes shared between the controls and IPF (Figure 7A). Of the three groups of genes analyzed, the first group contained genes that belong to the adaptation hypoxia response in controls (311); the analysis coincides with those that were made as a whole by condition and cell line. As these pathways are not found in IPF fibroblasts, they provide us with information that could help elucidate the mechanisms of IPF pathogenesis since, for some reason, they have been lost or blocked. In the second group, only 35 genes were found altered in both; these genes are involved in hypoxia and mesenchymal pathways. Finally, the genes that are induced by hypoxia and that are shared in different cells lines from IPF, Table 2 and Table 3 show the 19 DEGs, divided into up- and down-regulated; there is some discrepancy in the four down-regulated genes in our study because previous studies in IPF report them up-regulated, and this is the case with Coagulation factor III (F3), Hedgehog interacting protein (HHIP), Interleukin 6 (IL-6), and Stanniocalcin 1 (STC1) (Table 2).

Five genes were identified to be down-regulated that have not been studied in IPF, which are Cyclin G2 (CCNG2), Potassium Channel Tetramerization Domain Containing 16 (KCTD16), Basic helix-loop-helix family member e41 (BHLHE41), Syntaxin binding protein 6 (STXBP6), and Serpin family B member 7 (SERPINB7) (Table 2).

Nine genes were found up-regulated, four genes have been previously related to IPF: Endothelial PAS domain protein 1 (EPAS1), Transferrin receptor (TFRC), Endothelin receptor type A (EDNRA) and Periostin (POSTN). The other five genes have not been associated with IPF: the Alcohol dehydrogenase 1B (ADH1B), Protocadherin 18 (PCDH18), Homeobox A5 (HOXA5), Solute carrier family 14-member 1 (SLC14A1), and Ubiquitin specific peptidase 18 (USP18) (Table 3).

## 4. Discussion

Hypoxia is active in the lungs and fibroblasts of IPF patients. Several studies support the idea that hypoxia through the hypoxia-inducible transcription factors (HIF-1α and 2α) is a determining factor in the development and progression of fibrosis, participating in fibroblast activation and differentiation [4,5,6,8,40,41].

Most in vitro studies with lung fibroblasts from patients with IPF are performed under conditions other than the fibrotic microenvironment in a CO_2_ incubator, where oxygen concentration differ from concentration in the microenvironment typical of the fibroblast focus. Therefore, this work also provides information on the possible dysregulated pathways in the fibroblast foci of patients with IPF.

This exploratory and descriptive study was carried out to generate hypotheses about the possible signaling pathways involved in the hypoxia adaptation and the altered phenotype of IPF derived fibroblasts.

Therefore, we performed a gene expression analysis with a robust platform (Clariom D), where fibroblasts were analyzed under a hypoxia environment. This study shows that hypoxia induces essential changes in the expression of genes in controls related to metabolism, stress response, and epithelial differentiation, a modest response related to angiogenesis and regeneration in the case of IPF. Exclusive transcriptomic signature in IPF fibroblasts suggests a particular adaptation where EPAS1 (HIF-2α) plays a leading role.

As expected, the pathways shared between control fibroblasts and IPFs are related to hypoxia signaling. However, the intensity and complexity of the transcriptomic changes vary between the condition and the cell lines. On the whole, in the controls, there are more genes involved in pathways that are essential for cell maintenance in conditions of limited oxygenation, such as autophagy and metabolism. In fibroblasts derived from patients with IPF, a limited activation is observed, and similar results were observed with other stressors such as starvation and apoptosis inducers [42,43,44].

HIF-1α and 2α are known to have common target genes, yet it has been suggested that each isoform may exclusively regulate some genes [45,46,47,48]. It is also known that HIF-1α participates in a relevant way in acute hypoxia and that HIF-2α participates in chronic hypoxia [49,50]. So, we suggest that there probably exists a preadaptation in IPF cells that have been exposed to specific environments of chronic hypoxia.

As was foreseeable, the extracellular matrix-related genes are upregulated in hypoxia in both controls and IPF, so they are not exclusive to the specific transcriptional signature of IPF in hypoxia. However, the possibility remains that its increase is grater in IPF.

In the transcriptional signature that was observed in the genes exclusive to IPF, unexpected findings occurred, since no previous connection to hypoxia or IPF had been made such as the circadian rhythm and the differentiation of Th17 cells. In the case of the circadian rhythm pathway, it has been reported that oxygen through the activation of HIF1α, is a reset signal for circadian clocks [51].

Among the 19 DEGs found in the transcriptional signature, some genes such as EPAS1, POSTN, HOXA5, and HHIP are related to regeneration and development. We recently proposed that hypoxia signaling pathways are necessary for the context of lung regeneration. In addition, recapitulation of these pathways has been reported in pulmonary fibrosis [52]. However, if regeneration persists, hypoxia can activate feedback loops related to disease progression [7]. In animal models where the mechanisms involved in the regeneration process are studied, a prominent role of HIF-1α and HIF-2α was determined. It was observed that after an injury HIF-1α mRNA reaches a maximum peak on day 3 and then declines, while HIF-2α shows a high expression until day 13 and tends to decrease once the regeneration process has been carried out adequately [53,54]. The altered regeneration signaling observed in the IPF group could be a result of the differential HIF activity.

## 5. Conclusions

Pathways in IPF fibroblasts that are not shared with healthy cells are pathways that could help us begin to guide potential therapeutic targets. In the case of fibroblasts from patients with IPF, high levels of EPAS (HIF-2α) suggest that it plays an essential role in this chronic adaptation to hypoxia.

## Figures and Tables

**Figure 1 cells-11-03014-f001:**
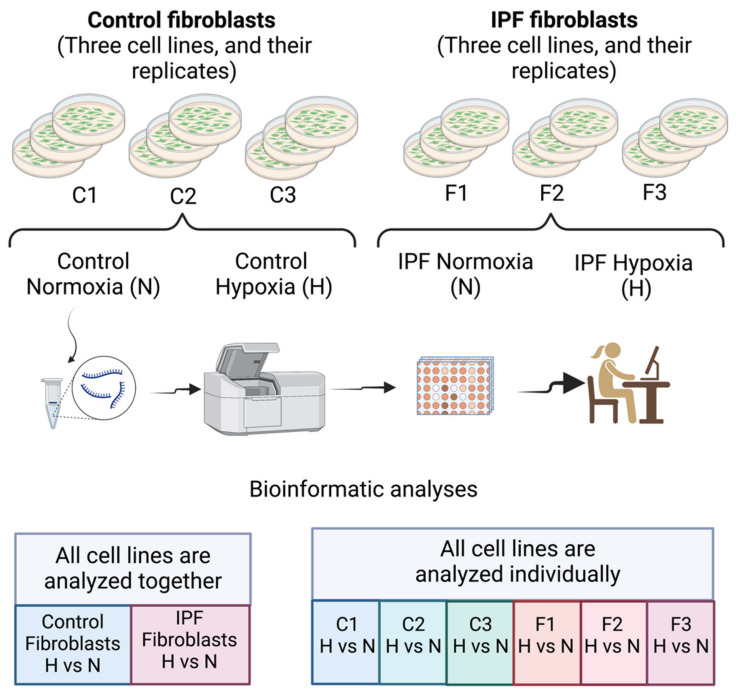
Design of the experiment. This figure was created with Biorender.

**Figure 2 cells-11-03014-f002:**
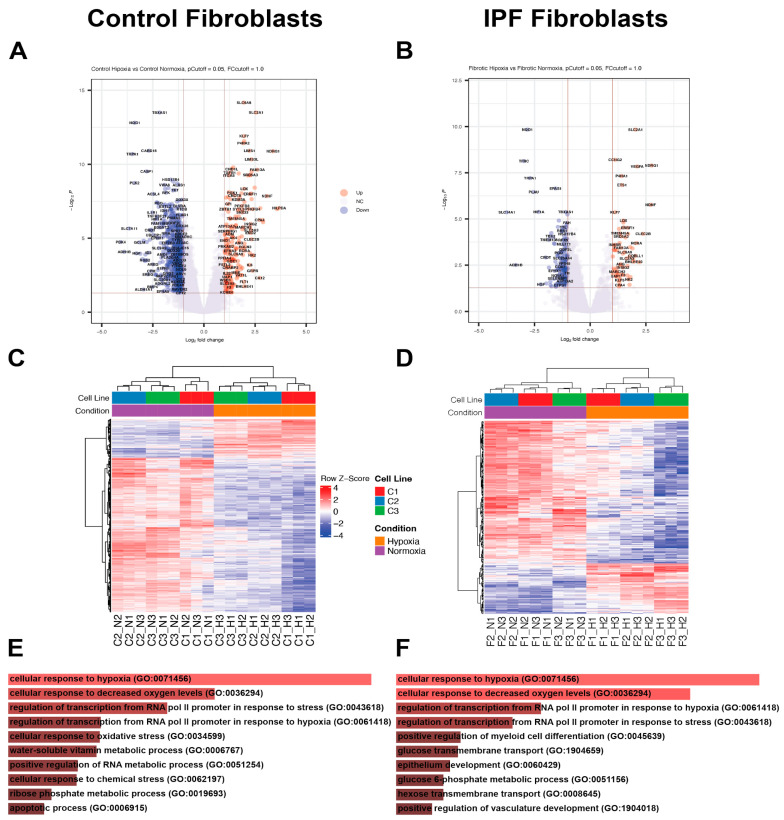
Response to hypoxia for 48 h in control and IPF fibroblasts. (**A**) Volcano plot (control fibroblasts). (**B**) Volcano plot (IPF fibroblasts). (**C**) Heatmap (control fibroblasts). (**D**) Heatmap (IPF fibroblasts). (**E**) GO gene ontology biological process (control fibroblasts). (**F**) GO gene ontology biological process (IPF fibroblasts). In subfigures (**A**–**D**), the colors red and blue correspond with upper and lower levels of expression.

**Figure 3 cells-11-03014-f003:**
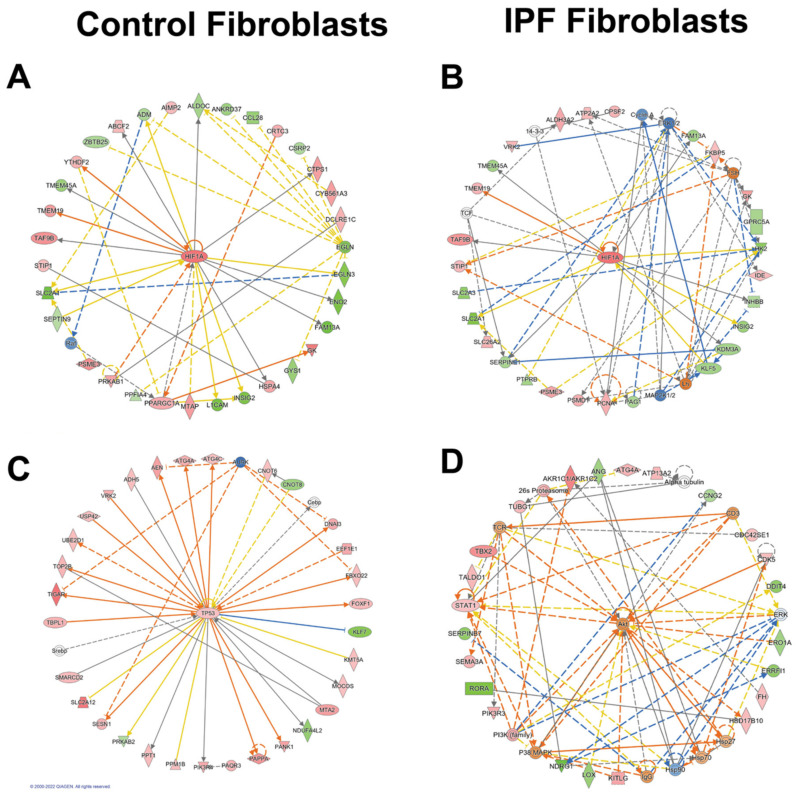
Networks in response to hypoxia in control and IPF fibroblasts. Radial diagrams of HIF1 surrounded by its targets (**A**) (control fibroblasts) and (**B**) (IPF fibroblasts). (**C**) Radial diagram of TP53 surrounded by its targets in control fibroblasts. (**D**) Radial diagram of AKT surrounded by its targets in IPF fibroblasts. Colored nodes refer to genes in our dataset (green down-regulated; red up-regulated). Uncolored nodes were not identified as differentially expressed in our experiment and were integrated into the computationally generated IPA networks. Arrows identify predicted relationships (orange leads to activation, blue leads to inhibition, yellow finds inconsistency with downstream molecules, and grey is no effect predicted).

**Figure 4 cells-11-03014-f004:**
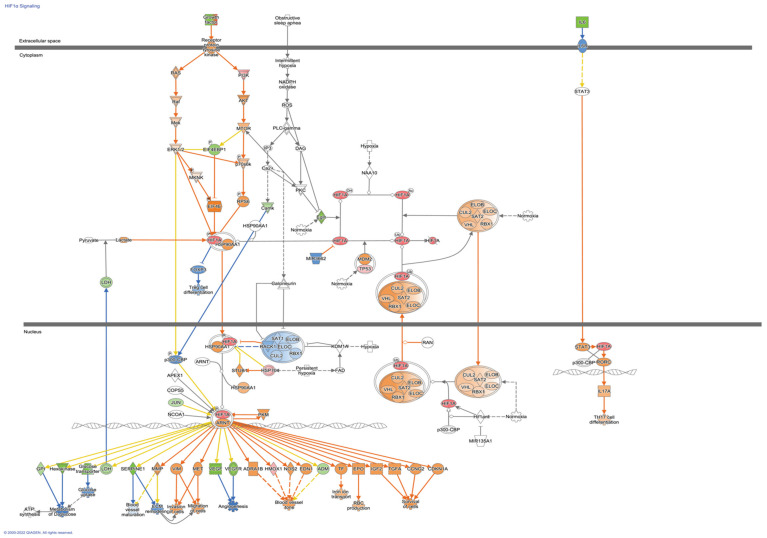
HIF-1α Signaling Pathways in control Fibroblasts. Colored nodes refer to differentially expressed genes found in our dataset control fibroblasts hypoxia vs. normoxia (green down-regulated; red up-regulated). Uncolored nodes were not identified as differentially expressed in our experiment and were integrated into the computationally generated IPA networks. This figure was created with Ingenuity Pathway Analysis, Version 2000–2022 QIAGEN. For further information about the symbols, please go through their web page: https://qiagen.secure.force.com/KnowledgeBase/articles/Knowledge/Legend.

**Figure 5 cells-11-03014-f005:**
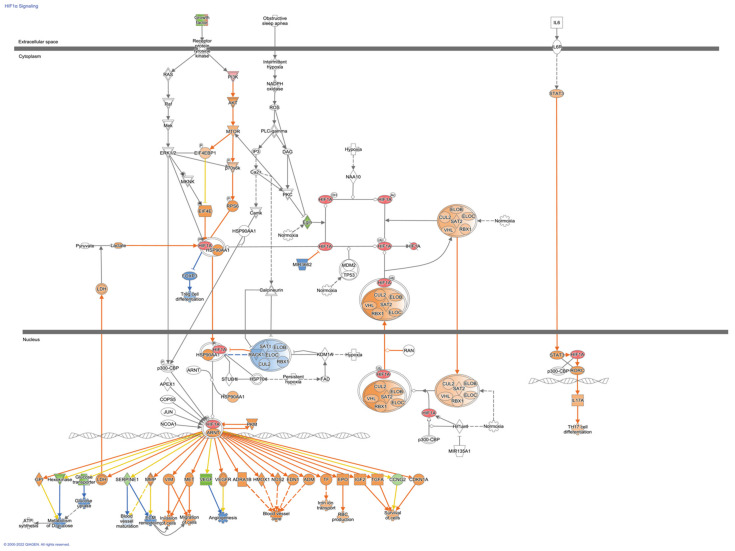
HIF1α Signaling Pathways in Fibroblasts from Idiopathic Pulmonary Fibrosis. Colored nodes refer to differentially expressed genes found in our dataset IPF fibroblasts hypoxia vs. normoxia (green down-regulated; red up-regulated). Uncolored nodes were not identified as differentially expressed in our experiment and were integrated into the computationally generated IPA networks. This figure was created with Ingenuity Pathway Analysis, Version 2000–2022 QIAGEN. For further information about the symbols, please go through their web page: https://qiagen.secure.force.com/KnowledgeBase/articles/Knowledge/Legend.

**Figure 6 cells-11-03014-f006:**
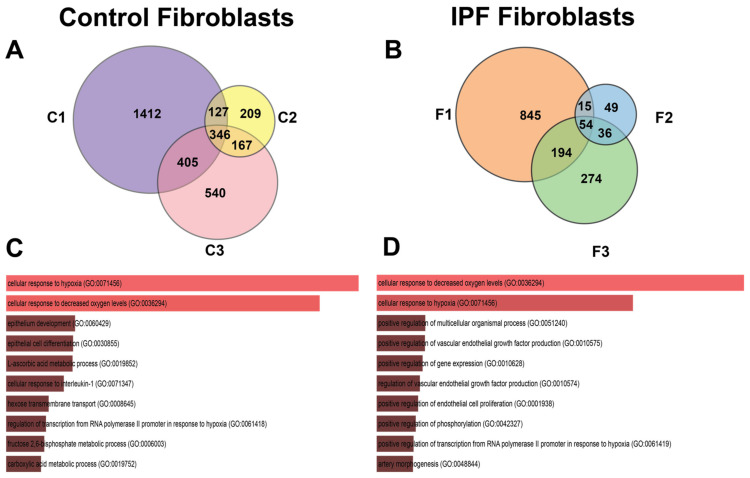
Venn diagrams of differential expressed genes in response to hypoxia in control and IPF fibroblasts (analyzing cell lines separately). In (**A**) Venn diagram (control fibroblasts), cell lines are represented with labels C1, C2, or C3. (**B**) Venn diagram (IPF fibroblasts), cell lines are represented with labels F1, F2, and F3. (**C**,**D**) represent the Gene Biological Process using EnrichR of the shared genes in their respective Venn diagram.

**Figure 7 cells-11-03014-f007:**
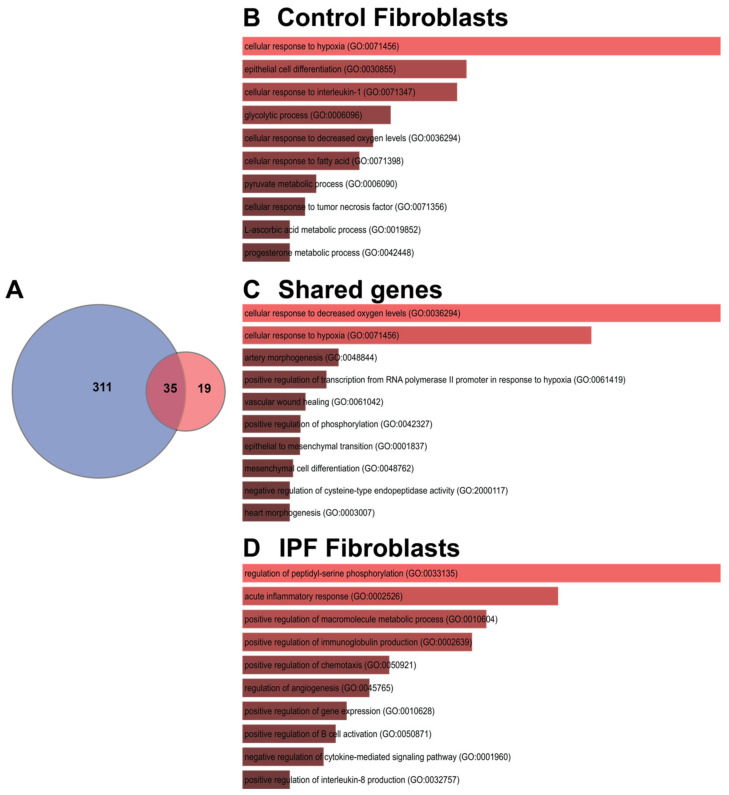
Shared genes in response to hypoxia. (**A**) Venn diagram represents the intersection of the differential expressed genes shared between cell lines in control fibroblasts or IPF fibroblasts; (**B**–**D**) represent the Gene Biological Process using EnrichR.

**Table 1 cells-11-03014-t001:** Demographic characteristics of the lung samples.

ID	Age	Condition	Passage Number	Primary or Commercial
C1	7 weeks	Healthy	12	ATCC CCL-215
C2	>50 years	Healthy	12	Primary
C3	79 years	Healthy	14	ATCC PCS-201-013
F1	>50 years	IPF	6	Primary
F2	>50 years	IPF	7	Primary
F3	58 years	IPF	7	Primary

All cells were cultured for 48 h in a hypoxic chamber at 1% O_2_.

**Table 2 cells-11-03014-t002:** List of up-regulated genes in lung fibroblasts from IPF patients in normoxia and hypoxia.

Gene	Also Known as	N	H	OS	Description	Models	References
F3	TF; AGT; CD142	NS	Up	Up	Coagulation factor III is a surface receptor, and it participates in the coagulation cascade	IPF-HLF	[27]
HHIP	HIP	NS	Up	Up	Hedgehog interacting protein is involved with Hedgehog signaling pathway in embryonic development	IPF-HLF	[28,29]
IL6	CDF; HGF; HSF; BSF2; IL-6; BSF-2; IFNB2; IFN-β-2	NS	Up	Up	Interleukin 6 encodes for a cytokine with inflammatory functions	IPF-HLF and BMM	[30,31]
STC1	STC	NS	Up	Up	Stanniocalcin 1 encodes for homodimeric glycoprotein with paracrine and autocrine functions	Increased in plasma of patients with IPF	[32]
DDIT4	Dig2; REDD1; REDD-1	NS	Up	-	DNA Damage Inducible Transcript 4. Gene related to response to virus, hypoxia, DNA damage, and tumor regulation	Expression associated with lncRNAs in IPF	[33]
CCNG2	-	NS	Up	-	Encodes for cyclin-G2 involved in cell cycle	NS-IPF	-
KCTD16	-	NS	Up	-	Potassium channel tetramerization domain containing 16 regulates GABA receptor signaling	NS-IPF	-
BHLHE41	DEC2; FNSS1; hDEC2; BHLHB3; SHARP1	NS	Up	-	Basic helix-loop-helix family member e41, involved in circadian rhythm and cell differentiation	NS-IPF	-
STXBP6	amisyn; HSPC156	Up	Up	-	Syntaxin binding protein 6 is involved in regulating SNARE complex formation	NS-IPF	-
SERPINB7	PPKN; TP55; MEGSIN	NS	Up	-	Serpin family B member 7 encodes for a protein that functions as a protease inhibitor	NS-IPF	-

N = Normoxia results in this study; H = Hypoxia results in this study; OS = Other Studies; IPF-HLF = IPF human lung fibroblast; BMM = Bleomycin mouse model; BAL = Bronchoalveolar lavage; NS-IPF = Not studied in idiopathic pulmonary fibrosis; Non-significant = NS. The search for the function of the genes was carried out using the gene card (https://www.genecards.org/). As for the studies reported in IPF, a basic search was performed in PubMed with the name of the gene or protein it encodes (https://pubmed.ncbi.nlm.nih.gov/), and only those that seemed relevant were considered.

**Table 3 cells-11-03014-t003:** List of down-regulated genes in lung fibroblasts from IPF patients in normoxia and hypoxia.

Gene	Also Known as	N	H	OS	Description	Models	References
EPAS1	HLF; MOP2; ECYT4; HIF2A; PASD2; bHLHe73	Down	Down	Up	Endothelial PAS domain protein 1 is a gene that encodes a transcription factor involved in signaling pathway in hypoxia	IPF-HLF	[9]
TFRC	CD71, IMD46, T9, TFR, TFR1, TR, TRFR, p90	Down	Down	Up	Transferrin receptor encodes a cell surface receptor associated with cellular iron uptake and is required for erythropoiesis	IPF-HLF, BMM and BAL of IPF patients	[34,35]
POSTN	PN; OSF2; OSF-2; PDLPOSTN	Down	Down	Up	Gen encodes for periostin protein with functions in tissue development and regeneration. Expression related to IPF progression	IPF-HLF and BMM	[36,37]
EDNRA	ETA; ET-A; ETAR; ETRA; MFDA; ETA-R; hET-AR	NS	Down	Up	Endothelin receptor type A encodes an endothelin-1 receptor with vasoconstriction properties	Primary rat alveolar type II cells	[38,39]
HOXA5	HOX1; HOX1C; HOX1.3	NS	Down	Up	Homeobox A5 encodes for transcription factors called homeobox genes spatially and temporally regulated during embryonic development	NS-IPF	-
PCDH18	PCDH68L	NS	Down	Down	Protocadherin 18 encodes for a protein member of the subfamily of cadherin superfamily related to cell–cell connections	NS-IPF	-
ADH1B	ADH2; HEL-S-117	NS	Down	-	Alcohol dehydrogenase 1B encodes for a protein member of the alcohol dehydrogenase family	NS-IPF	-
SLC14A1	JK; UT1; UTE; HUT11; Jk(a); Jk(b); RACH1; RACH2; UT-B1; HUT11A; HsT1341	NS	Down	-	Solute carrier family 14-member 1 encodes for a protein membrane transporter that mediates urea transport in erythrocytes	NS-IPF	-
USP18	ISG43; UBP43; PTORCH2	NS	Down	-	Ubiquitin specific peptidase 18 encodes for an enzyme that belongs to ubiquitin-specific proteases family	NS-IPF	

N = Normoxia results in this study; H = Hypoxia results in this study; OS = Other Studies; IPF-HLF = IPF human lung fibroblast; BMM = Bleomycin mouse model; BAL = Bronchoalveolar lavage; NS-IPF = Not studied in idiopathic pulmonary fibrosis; Non-significant = NS. The search for the function of the genes was carried out using the gene card (https://www.genecards.org/). As for the studies reported in IPF, a basic search was performed in PubMed with the name of the gene or protein it encodes (https://pubmed.ncbi.nlm.nih.gov/), and only those that appeared relevant were considered.

## Data Availability

Microarray data can be downloaded from https://figshare.com/projects/Hipoxia/141623.

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
