# Peer review of "Effect of Hypoxia in the Transcriptomic Profile of Lung Fibroblasts from Idiopathic Pulmonary Fibrosis"

_cells, 2022, doi:10.3390/cells11193014_

Round 1

Reviewer 1 Report

1. Please describe the hypoxia method briefly as readers may not want to download the cited paper for the method.

2.  All the methods need to be described properly. In its present form they are very short not fully understandable.

3. revealed a solid response to hypoxia in-what is a solid response-please use only scientific language

4. This phenomenon could be explained because IPF fibroblasts were previously exposed to this stimulus, sug- gesting preadaptation to this condition- cite

Author Response

Dear Reviewer 1

Thank you for allowing us to resubmit our manuscript. We have carefully reviewed your feedback and believe that we are addressing most of your concerns. This revised version has been reorganized according to your suggestions. We hope that with these changes, our manuscript can be accepted.

1.- Please describe the hypoxia method briefly as readers may not want to download the cited paper for the method.

Answer. Thanks for the comment; the method was described as requested.

Page 2, 3 and 4

3.- Revealed a solid response to hypoxia in-what is a solid response-please use only scientific language

Answer. Thank you for your valuable observation; the writing has been corrected according to your suggestion.

Page 5 line 364

4.- This phenomenon could be explained because IPF fibroblasts were previously exposed to this stimulus, suggesting preadaptation to this condition- cite

Answer. Thank you for your valuable observation; the corresponding reference has been added.

Page 5 line 368

Reviewer 2 Report

Please re-write large portions of the text as guided by comments. Introduction needs to be more detailed and tell readers a story about why you did the experiment at hand. 
We need to see more statistical analyses as well from the RNA seq of the cell lines because there is huge variability between samples. Also please mention exact number of days, passage number etc for each cell line culture. 

Avoid making sweeping conclusions from your data and may be undertake some validation experiments for the up and down regulated genes you end up focusing on. 

Author Response

Thank you for allowing us to resubmit our manuscript. We have carefully reviewed your feedback and believe that we are addressing most of your concerns. This revised version has been reorganized according to your suggestions. We hope that with these changes, our manuscript can be accepted.

We are very grateful for the corrections you sent us in the pdf file.

Please re-write large portions of the text as guided by comments. Introduction needs to be more detailed and tell readers a story about why you did the experiment at hand. 

Answer. A large part of the manuscript was rewritten and the language was revised; we appreciate this valuable observation as it allowed improvements in the work; we also improved the presentation of some figures

We consider adding the following text in the introduction

Hypoxia is a stress condition that influences cell fate by modifying numerous circuits. In this context, especially in the “fibroblast foci”, the main histopathological characteristic of these patients are the mechanisms of hypoxia adaptation that result in profibrotic feedback signaling which could perpetuate a fibrotic state [7]. For example, hypoxia and the hypoxia transcription factors (HIF-1α and 2α) are involved in the differentiation of myofibroblasts, extracellular matrix deposition, and alteration in the cell cycle [8,9]. In our previous work, IPF fibroblasts show a particular adaptation to hypoxia because they overexpress the alpha 1 and 2 subunits but not subunit 3 (a negative regulator) of HIF, suggesting a hyperactivation of this pathway even in the presence of oxygen [PMID: 31234835 9]. Although these transcription factors are altered, the impact of hypoxia on the tanscriptomic profile has not been determined.

Page 1 line 72 to 86

We need to see more statistical analyses as well from the RNA seq of the cell lines because there is huge variability between samples.

Answer. It is a correct comment since it is known that these cells behave in a very heterogeneous way, so the experimental design of this work was strongly considered since, if the literature is reviewed, few studies have used three primary lung lines from patients with FPI and three cell lines from healthy lungs (controls) and also in triplicate in each condition (normoxia and hypoxia); All this gives a total of 36 microarrays that is statistically solid, the variability is high and typical of these cells, however, thanks to the Venn diagram strategy, it was possible to purify the unique DEGs of the lung fibroblasts of patients with IPF in hypoxia.

We also consider improving the description of the Analysis of differential gene expression section.

Page 4 line 277 to 296

Also please mention exact number of days, passage number etc for each cell line culture. 

Answer. Thank you for your valuable observation, was corrected

Page 3, table 1

Avoid making sweeping conclusions from your data and may be undertake some validation experiments for the up and down regulated genes you end up focusing on.

Answer. Thank you for your valuable comment.

The experimental design of this work has an exploratory approach. It shows some strengths: we have used three primary lung lines from IPF patients and three cell lines from healthy lungs (controls). In addition, the experiments have been carried out in triplicate for each condition (normoxia and hypoxia) in a total of 36 microarrays; this supports the results and conclusions. Finally, we consider that validation would be part of the perspectives of the working group and other groups interested in this field.

We added a small paragraph in the discussion where we clarified this point about the experimental design.

"This exploratory and descriptive study was performed to generate hypotheses about potential signaling pathways involved in adaptation to hypoxia and altered phenotype of IPF-derived fibroblasts."

Discussion, line 570 to 572

Round 2

Reviewer 2 Report

I have added some comments. After minor revision, accept for publication.